# Induced Genetic Deletion of Cell Division Autoantigen 1 in Adulthood Attenuates Diabetes-Associated Renal Fibrosis

**DOI:** 10.3390/ijms26052022

**Published:** 2025-02-26

**Authors:** Pacific Huynh, Yuxin Yang, Hua Tian, Tieqiao Wu, Minling Huang, Jiali Tang, Aozhi Dai, Mark E. Cooper, Zhonglin Chai

**Affiliations:** Department of Diabetes, School of Translational Medicine, Monash University, Melbourne, VIC 3004, Australia; pacific.huynh@mssm.edu (P.H.); yangxx_123@126.com (Y.Y.); 18092090577@163.com (H.T.); tieqiao.wu@monash.edu (T.W.); minling.huang@monash.edu (M.H.); jiali.tang@monash.edu (J.T.); aozhi.dai@monash.edu (A.D.); mark.cooper@monash.edu (M.E.C.)

**Keywords:** diabetic kidney disease, TGFβ, CDA1, renal fibrosis, inducible knockout

## Abstract

Cell Division Autoantigen 1 (CDA1) has been shown to play a role in enhancing transforming growth factor beta (TGFβ) signaling, leading to fibrosis in diabetic kidney disease (DKD) using mouse strains with global CDA1 gene deletion. In these models, diabetes has been induced, leading to DKD in the absence of CDA1. It is still unknown whether inhibition of CDA1 activity after onset of diabetes in the presence of CDA1 can attenuate renal fibrosis in vivo. Thus, we examined the effect of inducing genetic deletion of CDA1 in adulthood in mice using a tamoxifen-activated estrogen receptor fused cyclization recombinase (ERCre)-Locus of cross-over in P1 (LoxP) system. Male mice at 6–8 weeks of age were rendered diabetic with streptozotocin (STZ) or injected with buffer alone to serve as non-diabetic controls. Five weeks later, genetic deletion of CDA1 was induced by tamoxifen administration in CDA1Flox/ERCre mice, with mice injected with vehicle to serve as CDA1 wildtype controls. Kidney tissues were analyzed 5 weeks after deletion of CDA1. Tamoxifen administration reduced CDA1 gene expression by ~80% in CDA1Flox/ERCre mice. Renal levels of phosphorylated Smad3 and expression of profibrotic genes as well as accumulation of extracellular matrix proteins (ECMs) such as collagens III and IV were increased in diabetic mice, and induced deletion of CDA1 led to attenuation of these parameters. Therefore, targeting CDA1 after onset of diabetes in mice where CDA1 was initially expressed is able to attenuate diabetes-associated renal injury, providing the impetus to target this pathway in order to reduce diabetic kidney disease.

## 1. Introduction

Fibrosis is the common final pathological event in chronic kidney diseases, such as in diabetic kidney disease. The severity of fibrotic lesions in the kidney, characterized by the excessive accumulation of extracellular matrix components, generally correlates with a decline in renal function, as well as in progression towards kidney failure [1,2,3]. One of the key factors associated with fibrosis is enhanced TGFβ signaling in response to injury [4]. While previous studies have demonstrated that inhibition of TGFβ activity attenuates the development of renal fibrosis in various animal models of renal fibrosis [5,6,7], due to its physiological importance, complete inhibition of the TGFβ signaling pathway is associated with deleterious side-effects, as has been shown in mice with genetic deletion of the TGF-β1 gene [8,9,10].

Cell Division Autoantigen 1 (CDA1), encoded by the gene *TSPYL2*, is also known as Differentially Expressed Nucleolar TGF-beta1 Target (DENTT) and Cask-interacting Nucleosome Assembly Protein (CINAP) [11,12,13]. It is a nuclear phosphoprotein and a member of the SET/NAP/TSPY protein family. CDA1 has been implicated in multiple biological processes, such as cell proliferation [11,14,15], DNA damage response [14,16,17,18], cellular senescence [19], tumor suppression [15,20,21], epigenetic modifications [17,22,23,24], and neurologic biology [13,22,25,26,27,28]. We have extensively characterized CDA1 to show that it plays a critical role in the development of fibrosis in rodent models of diabetic nephropathy and diabetes-associated vascular fibrosis [29,30,31,32,33]. Indeed, renal biopsy samples from human patients with renal fibrosis were associated with elevated levels of CDA1 protein [31]. The pathological role of CDA1 in renal fibrosis has been attributed to its ability to enhance the profibrotic actions of TGFβ signaling, with in vitro studies revealing a synergistic relationship between TGFβ and CDA1 [12,15,20,29,30]. The inhibition or absence of CDA1 has been demonstrated to attenuate the development of fibrosis in vitro, with silencing of CDA1 expression leading to reduced TGFβ-stimulated activation of Smad3, and consequently reduced profibrotic gene expression [29,30]. CDA1 appears to influence the TGFβ signaling pathway by increasing the expression level of the TGFβ type I receptor (TβRI), the kinase receptor directly responsible for phosphorylation and activation of the signaling molecules Smad2/3 [30]. The TGFβ/Smad3 axis has been shown to promote renal fibrosis, whereas Smad2 is considered to counteract the profibrotic activity of Smad3 [34]. Indeed, this action of CDA1 has been shown to be specifically inhibited by the TβRI inhibitor, SB431542, as well as by the Smad3 inhibitor, SIS3 [30]. These findings were validated in vivo, where global genetic deletion of CDA1 attenuated renal fibrosis development in a mouse model of diabetic nephropathy [31].

With these data supporting CDA1 being a potential therapeutic target to attenuate the development of renal fibrosis in DKD, pharmacological inhibitors of CDA1 are currently at a developmental/preclinical stage [35]. A pharmacological inhibitor would act in the context of disease initiation and progression in the presence of CDA1. Thus, we experimentally reproduced this context using an inducible CDA1 knockout (KO) mouse strain, the CDA1Flox/ERCre mouse. In this model, diabetes was able to be induced, leading to diabetes-associated renal injury in the presence of CDA1. In this study, we demonstrate that the deletion of CDA1 at week 5 after induced diabetes, mimicking the therapeutic inhibition of CDA1 once renal injury is developing, led to an attenuation in diabetes-associated renal fibrosis.

## 2. Results

### 2.1. Tamoxifen Treatment Significantly Reduces CDA1 Gene Expression in CDA1Flox/ERCre Mice

Genotyping using polymerase chain reaction (PCR) analysis on genomic DNA isolated from mouse tissues showed that the genetic deletion of CDA1 was induced by tamoxifen treatment at 1 mg/day in CDA1Flox/ERCre mice, but not in CDA1Flox nor in ERCre mice, as designed (Figure A1). We then examined how the induced genomic deletion of the CDA1 gene affected the mRNA levels of CDA1 in the kidney, the transcripts that produce the functional CDA1 protein. Reverse-Transcription PCR (RT-PCR) was performed to analyze the structure of the CDA1 gene transcript using cDNA templates derived from the total RNA isolated from the mouse kidney tissues (Figure 1a). Using a CDA1 wildtype mouse and a global CDA1 knockout mouse as known genotype controls, a knockout band of ~389 bp was produced from the CDA1Flox/ERCre mice treated with tamoxifen for either 3 or 6 days, which, as expected, was not seen in CDA1Flox mice without the ERCre transgene (Figure 1a). In these mice, a wildtype band of ~825 bp was visible, indicating there were still some renal cells where the CDA1 gene was not deleted. It is worth noting that CDA1Flox/ERCre mice treated with vehicle (no tamoxifen) for 3 or 6 days had a very faint, almost invisible, band equivalent to the knockout band of ~389 bp, suggesting that the background level of the CDA1 gene knockout transcript was minimal (Figure 1a). The relative quantities of the wild type CDA1 transcripts were measured by quantitative real-time RT-PCR, which showed that 3 daily injections of tamoxifen at 1 mg/day reduced CDA1 mRNA levels by ~50%, which were further reduced by ~74% by 3 daily injections of tamoxifen at 1.5 mg/day (Figure 1b). Based on these results, tamoxifen administration at 1.5 mg per day for 3 consecutive days was used to delete CDA1 in the diabetic kidney disease model in CDA1Flox/ERCre mice in order to examine the effect of targeted deletion of CDA1 on the progression of renal injury after establishment of diabetic kidney disease.

### 2.2. Renal CDA1 Gene Expression Was Persistently Reduced 5 Weeks After Tamoxifen Treatment in Control and Diabetic CDA1Flox/ERCre Mice

Both CDA1Flox and CDA1Flox/ERCre mice were rendered diabetic by STZ injections or injected with buffer alone to serve as non-diabetic controls (Figure 2a). Four weeks later, both CDA1Flox and CDA1Flox/ERCre mice injected with STZ exhibited typical diabetes-associated changes in metabolic parameters, including lower body weight, increased food intake, polydipsia and polyuria (Table 1). All mice were randomly allocated to receive either tamoxifen (3 daily injections of 1.5 mg tamoxifen per day) or vehicle at 5 weeks after STZ injections. Five weeks later, at the 10-week endpoint, CDA1Flox mice that did not possess the ERCre transgene had similar levels of renal CDA1 gene expression, regardless of whether they were administered vehicle or tamoxifen (Figure 2b). There was a tendency towards a diabetes-associated elevation in renal CDA1 gene expression in these mice, albeit the difference was not statistically significant in this study. Tamoxifen administration led to a ~75–80% reduction in renal CDA1 gene expression in CDA1Flox/ERCre mice, compared to vehicle-treated animals, regardless of diabetic status (Figure 2b, *p* < 0.001 compared to vehicle-treated mice). Genetic deletion of CDA1 was also confirmed by analyzing the renal CDA1 transcripts by PCR using cDNA as templates. As shown in Figure 2c, a knockout (KO) band at ~386 bp, reflecting the deletion of exons 2–5 of the CDA1 transcript, was present in CDA1Flox/ERCre mice receiving tamoxifen (Figure 2c), whereas the wild type CDA1 (WT) band at ~817 bp was visible, albeit in a faint intensity, indicating a small number of kidney cells still had the intact CDA1 gene. As expected, in the CDA1Flox/ERCre mice receiving vehicle treatment without tamoxifen, the WT band was the major PCR product, with a minimum amount of the KO band produced, which was almost invisible. In the CDA1Flox mice which had no ERCre gene, only the WT band was produced. These results from the control groups further confirmed that the CDA1 deletion seen in the kidney of CDA1Flox/ERCre mice occurred as a result of tamoxifen mediated activation of transgenic ERCre leading to the specific deletion of exons 2–5 of the CDA1 gene. Thus, these results demonstrated that the CDA1Flox/ERCre mice were exposed to diabetes for 5 weeks in the presence of CDA1 and then were followed for a subsequent period of 5 weeks in the absence of CDA1.

### 2.3. Induced CDA1 Deficiency Does Not Affect Diabetes-Associated Metabolic Parameters

This study was designed to examine whether the deletion of CDA1 after renal disease had been established was able to protect the kidney from further injury by diabetes-associated stimuli. Therefore, we examined the metabolic parameters at the end point to ensure that the animals had remained diabetic even after the CDA1 gene had been deleted. As shown in Table 2, diabetic CDA1Flox and CDA1Flox/ERCre mice, receiving either vehicle or tamoxifen, exhibited an expected lower body weight, increased food intake, polydipsia and polyuria when compared to their non-diabetic counterparts. An increase in hemoglobin A1c (HbA1c) levels in these diabetic mice in comparison to their non-diabetic controls (Table 2) was confirmed, with no difference between groups with or without renal CDA1 gene deletion.

### 2.4. Diabetes-Associated Kidney Hypertrophy and Tubular Injury Are Attenuated as a Result of CDA1 Deficiency

Diabetic mice from both strains, treated with either vehicle or tamoxifen, showed a significant increase in the kidney weight/body weight ratio (*p* < 0.001) (Figure 3a), reflecting diabetes-induced kidney hypertrophy in these mice. In CDA1Flox/ERCre mice receiving tamoxifen that had the CDA1 gene deleted at week 5 of diabetes, kidney hypertrophy was attenuated when compared to mice receiving vehicle where the CDA1 gene was intact (*p* < 0.05) (Figure 3a). Urinary KIM1 excretion, a tubular injury marker, was significantly increased in diabetic CDA1Flox/ERCre mice receiving vehicle where the CDA1 gene is intact (*p* < 0.001) (Figure 3b). An increase in this parameter was attenuated as a result of CDA1 deletion in the CDA1Flox/ERCre mice receiving tamoxifen (*p* < 0.05) (Figure 3b).

### 2.5. Diabetes-Associated Increases in Renal Expression of Profibrotic and Proinflammatory Molecules Are Attenuated in Mice with Induced CDA1 Deficiency

Renal injury was examined in mice with STZ-induced diabetes for 10 weeks in the presence of CDA1, as well as in those where the CDA1 gene was deleted for the last 5 weeks of diabetes. As expected, profibrotic genes such as collagens I, III, fibronectin and matrix metalloproteinase 2 (MMP2), as well as proinflammatory genes such as interleukin 6 (IL6) and osteopontin (OPN), were increased in diabetic CDA1Flox/ERCre mice receiving vehicle, which had the presence of the CDA1 gene for the entire 10-week period of diabetes (Figure 4). In the mice who received tamoxifen treatment leading to CDA1 gene deletion, the diabetes-associated increases in the expression of these genes were significantly attenuated, showing no statistical difference when compared to their non-diabetic counterparts (Figure 4).

The accumulation of extracellular matrix molecules, such as collagens, was examined by immunohistochemical staining in the kidney sections of the CDA1Flox/ERCre mice (Figure 5a). Fibrillar collagen III accumulation was found to be elevated more than 5-fold in the diabetic mice receiving vehicle in the presence of CDA1 when compared to the non-diabetic counterparts (Figure 5b). This diabetes-associated collagen accumulation was significantly attenuated, to a level similar to that seen in non-diabetic counterparts, as a result of CDA1 deletion (*p* > 0.05) (Figure 5b). A similar pattern was seen for collagen IV staining (Figure 6a), which was increased by ~3-fold in diabetic mice in the presence of CDA1. The induced deletion of CDA1 attenuated this effect, leading to a similar staining of collagen IV, as seen in the non-diabetic controls (Figure 6b).

### 2.6. Diabetes-Associated Increases in Renal Levels of Phosphorylated Smad3 Are Attenuated in Mice with Induced CDA1 Deficiency

In order to examine whether the above changes in renal fibrosis were due to the regulatory effects of TGFβ signaling in the mouse kidney, renal levels of phosphorylated Smad3 in these mice were analyzed by immunohistochemical staining (Figure 7a). Indeed, renal levels of phosphorylated Smad3 were increased by >2.5-fold in diabetic mice treated with vehicle (no tamoxifen), which were reduced to levels similar to those in the non-diabetic mice as a result of inducing deletion of CDA1, as seen in the group of diabetic mice treated with tamoxifen (Figure 7b).

## 3. Discussion

We have previously shown that CDA1 is upregulated in the kidney biopsy tissues from subjects with DKD as well as in non-diabetic fibrotic kidney disease [31]. The role of CDA1 in kidney fibrosis, the pathological hallmark of DKD, has been characterized previously in a series of in vitro and in vivo experiments [29,30,31]. These studies, including the use of global CDA1 knockout mice, demonstrate that CDA1 is a potential molecular target to retard DKD. In the clinical setting, clinical studies emphasize that DKD is a disorder where underlying structural injury can be reversed [36], thereby preventing the costly burden of ESRD, specifically dialysis and/or transplantation with its associated high mortality. Our recent identification of a putative receptor for CDA1, CDA1 binding protein 1 (CDA1BP1), provides the impetus to consider the CDA1/CDA1BP1 axis as a potential target for renoprotection. Indeed, it was shown that this axis could be safely and efficaciously targeted by a pharmacological approach in an animal model of DKD [35].

The previous studies to characterize and validate CDA1 as a target in vivo were performed using global CDA1 KO mice [31]. In those mice, CDA1 is absent during animal development and growth, representing a valuable model to study the preventive effect of CDA1 inhibition. However, in the clinical setting, diabetes and the consequent development of DKD occur in the presence of CDA1. Therefore, a more relevant approach for a clinical perspective would be to inhibit CDA1 once there is already some renal injury. Such an approach could involve pharmacological therapy, where it would be feasible to inhibit CDA1’s pathological activity to treat established DKD, thus slowing down, arresting or reversing the progression of established DKD. Therefore, an appropriate animal model is required to mimic such a clinical setting, where DKD is allowed to develop in the presence of CDA1, and then the CDA1 gene can be deleted at a selected time point after DKD is established.

The CDA1Flox/ERCre mouse using a modified Cre-LoxP system [37] was generated in this study by crossing the CDA1Flox mouse [31] with the ERCre transgenic mouse [38,39], where deletion of exons 2–5 of the CDA1 gene flanked by the LoxP sites is inducible upon administration of tamoxifen. Since ERCre is expressed globally, tamoxifen administration should lead to global deletion of CDA1 at the selected time of induction, a model well representing the systemic targeting of CDA1 by a pharmacological approach. Both ERCre and CDA1Flox mice were used as genotype controls, which were confirmed to have a genotype equivalent to wild type CDA1 (WT) with or without administration of tamoxifen. The CDA1Flox/ERCre mice were shown to have the CDA1 gene efficiently deleted upon administration of tamoxifen, leading to a remarkable reduction of renal CDA1 mRNA levels in both non-diabetic and diabetic mice (Figure 2). Therefore, this model is appropriate to be used to examine the effect of targeting CDA1 on the progression of the established diabetic kidney disease. Indeed, induced inactivation of CDA1 for 5 weeks by a genetic approach in this model led to attenuation of renal parameters relevant to DKD. This is consistent with our previous findings using a pharmacological approach where STZ-diabetic mice were treated with a prototype inhibitor of CDA1, CHA-061, for 5 weeks during weeks 6–10 after diabetes induction [35]. However, CDA1 protein expression levels were not examined in this study due to lack of specific antibodies to detect mouse CDA1 in kidney tissues, which is considered to be a limitation of this study. The polyclonal antibodies we generated many years ago were raised against a human CDA1 immunogen [11], which did not work on mouse kidney tissues.

The concurrent activation of proinflammatory and TGFβ signaling/ECM profibrotic pathways is a key feature of DKD in humans [40]. In rodents, these pathways are also activated as a result of diabetes, and thus have been useful models to study the pathophysiology of DKD, as well as to identify and validate potential drug targets to retard DKD [41,42]. With animal models not developing extensive end-stage kidney disease [43], this study has focused on the gene expression of key molecules involved in profibrotic and proinflammatory pathways, as well as extracellular matrix accumulation, which were found to be elevated in the diabetic mice in comparison with their non-diabetic counterparts. The diabetes-associated elevation of expression of these genes was attenuated as a result of the absence of CDA1 for the last 5 weeks in this study, as a result of the CDA1 gene being deleted 5 weeks after induction of STZ-diabetes. Furthermore, renal accumulation of collagen III, a scarring collagen, and collagen IV, a structural basement membrane collagen, were both increased in diabetic mice and attenuated in diabetic mice with deletion of the CDA1 gene. TGFβ is a profibrotic growth factor, playing a key pathological role in DKD. Consistent with our previous finding using global CDA1 knockout mice [31], TGFβ signaling assessed by the renal level of phosphorylated Smad3 in this study was significantly enhanced in diabetic mice, which was reduced to a physiological level as a result of induced CDA1 deletion. Smad3 is phosphorylated by the TGFβ type I receptor upon activation as a result of TGFβ ligand binding and then translocated to the nucleus, where it binds to the promoters and regulates the expression of TGFβ target genes. The expression levels of the TGFβ target genes examined in this study, such as collagens I, III [44], fibronectin [45] and matrix metalloproteinase 2 (MMP2) [46] were changed in a similar way to the levels of phosphorylated Smad3 (Figure 4 and Figure 7), supporting the role of CDA1 in influencing the profibrotic TGFβ/Smad3 pathway. Despite the attenuation of TGFβ signaling and renal fibrosis, there was no statistically significant difference in albuminuria seen between the diabetic groups with and without CDA1. This was, at least in part, due to a large variation of data and a relatively small group size. Furthermore, it is known that mice on a C57 background do not develop robust albuminuria. Furthermore, targeting TGFβ signaling may not necessarily affect albuminuria, as has been reported previously in numerous mouse models of DKD [31,47,48]. Indeed, CDA1 gene expression was not robustly increased in the diabetic kidney in this study where CDA1Flox mice and ERCre mice on C57 backgrounds were used. This is probably reflected by some parameters failing to show statistically significant differences between CDA1 WT and KO diabetic mice. Such an example is the mRNA levels of collagen III (*p* = 0.06) (Figure 4), although a clear tendency towards attenuation in the diabetic CDA1 KO mice was observed. Furthermore, the collagen III protein levels were shown to be significantly different in that group (*p* < 0.05) (Figure 5). Probably, a larger group size should be considered to increase the statistical power in future studies if one uses mice on a C57 background. Taken together, our results demonstrate that targeting CDA1 in established DKD is able to reduce TGFβ signaling, leading to reduced gene expression and protein accumulation of molecules associated with fibrosis in the kidney, and can potentially retard DKD progression.

## 4. Materials and Methods

### 4.1. Generation of CDA1Flox/ERCre Mice

CDA1Flox/ERCre mice were generated by crossing two mouse strains, the previously established CDA1Flox and ERCre strains, both on a C57 background. The CDA1Flox mouse has LoxP sites inserted to flank exons 2–5 of the CDA1 encoding gene *Tspyl2* [31]. The ERCre mouse, also known as the CAGGCre-ER^TM^ mouse, expresses the recombinant enzyme, ERCre, under the global CAGG promoter [49]. The recombinant ERCre is usually localized to cytoplasm, unable to interact with the target gene within the nucleus. Tamoxifen can be administered to the mice with ERCre, and the ERCre can be “activated” by translocation to the nucleus as a result of binding with tamoxifen, where it cleaves the genomic DNA at the LoxP sites, leading to knockout of the “floxed” gene. The CDA1Flox/ERCre mice with the CDA1 gene floxed and transgenic ERCre expression were used to achieve the inducible deletion of CDA1 by tamoxifen at a selective time point in adulthood, such as during the development of induced diabetic kidney disease. The littermate CDA1Flox (having no ERCre transgene) and ERCre (the CDA1 gene not floxed) mice were used as genotype controls, where the CDA1 gene was not deleted with or without tamoxifen administration.

### 4.2. Optimization of Tamoxifen Dose

Male CDA1Flox/ERCre mice (10–12 weeks of age) were injected intraperitoneally with tamoxifen (Sigma-Aldrich, St. Louis, MO, USA; Cat. No. T5648) dissolved in corn oil (Sigma-Aldrich; Cat. No. C8267) at either 1.0 or 1.5 mg per mouse per day for 3–6 consecutive days. Littermate CDA1Flox mice without the transgenic ERCre served as genotype controls. Mice designated as vehicle controls were injected with corn oil alone. Mice were culled 3–14 days after their final injection, and organs were collected for genotyping and gene expression analysis to determine the efficiency of CDA1 deletion.

### 4.3. Induction of Diabetes and Genetic Deletion of CDA1

CDA1Flox/ERCre male mice (6–8 weeks of age) were rendered diabetic by injections of STZ at 55 mg/kg per day for 5 consecutive days, as previously described [50], with littermate CDA1Flox and ERCre mice serving as genotype controls. Animals which served as non-diabetic controls were injected with citrate buffer alone. Blood glucose levels and body weight of the mice were monitored weekly to confirm the diabetic status. Mice were housed in a 12 h light/dark cycle with free access to water and standard mouse chow. At 5 weeks after STZ injections, tamoxifen (1.5 mg/mouse/day for 3 consecutive days) was administered intraperitoneally; the dose, chosen based on the findings from the pilot study, represents an intervention to suppress CDA1 activity. Mice injected with vehicle (corn oil) served as controls. Mice were euthanized using Barbiturate overdose (160 mg/kg sodium pentobarbital) at 10 weeks after diabetes was induced, and tissues were collected for analysis. Animal studies were performed at the Alfred Medical Research and Education Precinct (AMREP) Animal Services and approved by the AMREP Animal Ethics Committee (Approval Number: E/1271/2012/B) according to international guidelines, including the “Australian Code of Practice for the Care and Use of Animals for Scientific Purpose, 8th edition” (National Health and Medical Council of Australia, 2013).

### 4.4. Measurements of Metabolic Parameters

One week prior to tamoxifen administration and euthanasia, mice were placed individually into metabolic cages (Iffa Credo, L’Arbresle, France) to measure food and water intake, as well as urine output, over a 24 h period. Blood and urine were collected from these mice for further biochemical analysis. Glycated hemoglobin was measured from blood samples using the Cobas b 101 POC system (Roche Diagnostics, Basel, Switzerland) according to the manufacturer’s protocol. Urinary levels of kidney injury molecule-1 (KIM1) were measured using a mouse HAVcr-1 ELISA kit (EIAab, Wuhan, China; Cat. No. E0785m) in accordance with the protocol provided by the manufacturer.

### 4.5. Genotyping CDA1Flox/ERCre Mice

Genotyping of mice was carried out by standard PCR on genomic DNA isolated from mouse tissue to determine the presence or absence of target genes (sequences of primers used are listed in Table A1).

### 4.6. Quantitative Real-Time Polymerase Chain Reaction (qRT-PCR)

The extraction of RNA from tissue, cDNA synthesis and subsequent analysis via qRT-PCR were performed as described previously (sequences of primers and probes used are listed in Table A2) [51,52]. The gene-specific signals were normalized to signals of internal 18S ribosomal-RNA amplification utilizing the comparative threshold cycle (CT) method [53].

### 4.7. Histological Analysis

Paraffin sections of mouse kidney were stained with specific antibodies for phosphorylated Smad3, collagens III and IV, as described previously [31]. Twenty images of the renal cortex of each section were photographed at 200× magnification in a blinded manner. The positive staining was quantified using Image-Pro Plus 7.0 software (Media Cybernetics, Bethesda, MD, USA). The histogram-based mode of color thresholding was used to digitally analyze histological staining and determine the percentage area of positive staining in each image. The three red–green–blue parameters were manually gated between 0 and 255 to exclusively encompass the color of a specific stain, with brown for positive immunohistochemical staining. Accuracy of these parameters was confirmed with negative control staining.

### 4.8. Statistical Analysis

Statistical analysis was performed using GraphPad Prism 7 (GraphPad Software, La Jolla, CA, USA). All data were expressed in mean ± SEM. Differences in the mean among groups were analyzed using two-way ANOVA followed by multiple comparisons between groups using the Holm–Sidak post hoc test or using the Mann–Whitney test, as specified. A difference of *p* < 0.05 was considered to be statistically significant.

## Figures and Tables

**Figure 1 ijms-26-02022-f001:**
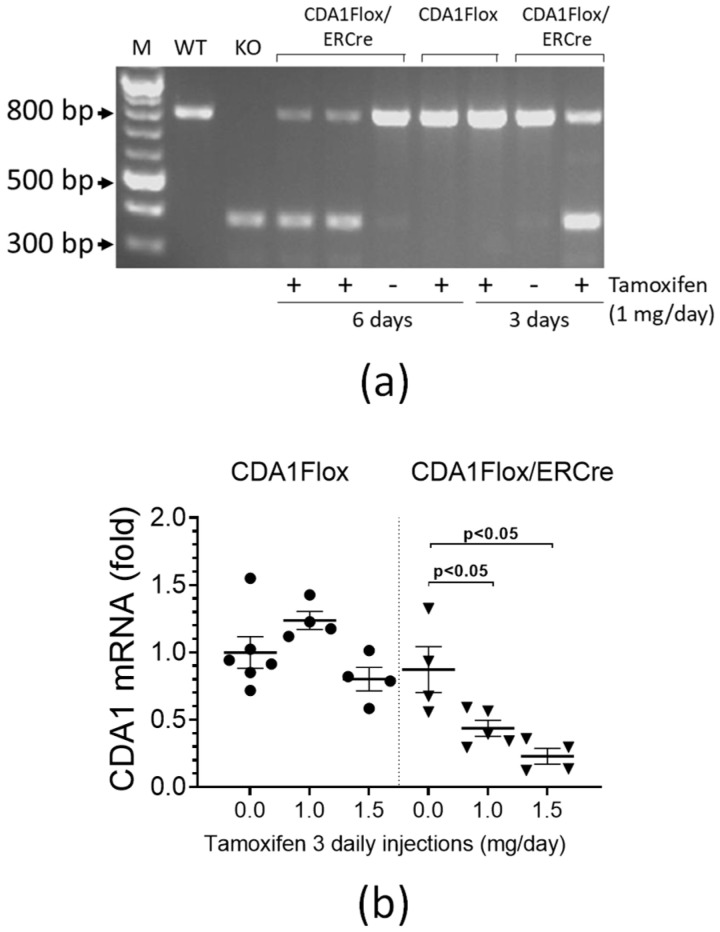
Renal CDA1 gene expression is remarkably reduced as a result of tamoxifen-induced specific deletion of *Tspyl2* gene encoding CDA1 in CDA1Flox/ERCre mice. (**a**) RT-PCR was performed on cDNA templates synthesized from total RNA extracted from mouse kidneys, using RT-PCR primers (Table A1) designed to detect the full length CDA1 transcript by producing an 817 bp fragment and the CDA1 KO gene transcript by producing a 386 bp fragment. (**b**) Renal CDA1 mRNA expression levels (fold change) were determined by quantitative real-time PCR (qRT-PCR) from CDA1Flox and CDA1Flox/ERCre mice receiving 3 daily intraperitoneal injections of vehicle (0.0 mg/day tamoxifen) or tamoxifen at 1.0 or 1.5 mg/day. Relative mRNA expression levels are shown from individual mice in each experimental group, and the group mean ± SEM and *p* values between specified groups are also shown.

**Figure 2 ijms-26-02022-f002:**
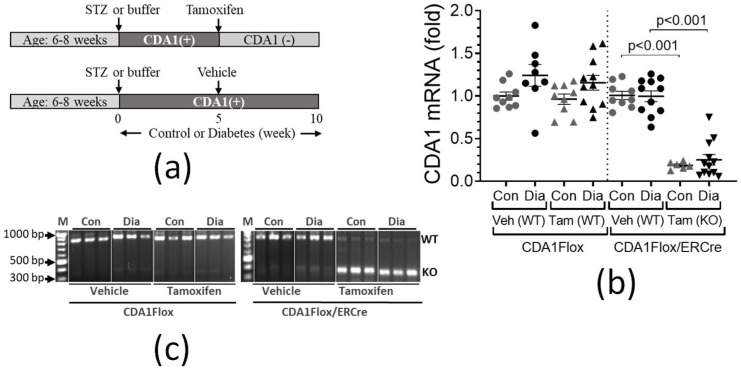
Characterization of mouse model of STZ-diabetes with inducible deletion of CDA1 gene. (**a**) Schema showing the timeline of the mouse model. Male mice of CDA1Flox and CDA1Flox/ERCre strains at the age of 6-8 weeks were peritoneally injected daily with streptozotocin (STZ) at 55 mg/Kg for 5 consecutive days to be rendered diabetic or with buffer alone to serve as non-diabetic controls. Five weeks later, both diabetic and non-diabetic mice were randomly allocated to receive tamoxifen treatment at 1.5 mg/mouse via 3 daily intraperitoneal injections to induce CDA1 gene deletion (CDA1(−)), which is designed to occur only in the CDA1Flox/ERCre mice (upper panel) but not in CDA1Flox mice, or to receive vehicle treatment to leave the CDA1 gene intact (CDA1(+)) (lower panel). Five weeks after treatment with tamoxifen or vehicle, the mice were individually placed in a metabolic cage to collect urine samples for 24 h to determine kidney function and were killed to collect kidney tissues for analysis. (**b**) Renal CDA1 mRNA expression levels (fold change) were determined by qRT-PCR, using gene specific primers and probes (see sequences in Table A2) designed to detect intact CDA1 transcripts. Relative mRNA expression levels are shown from individual mice in each experimental group, and the group mean ± SEM and *p* values between specified groups are also shown. (**c**) RT-PCR was performed using renal cDNA from each group (*n* = 3) as templates to determine deletion of the CDA1 gene. PCR products amplified from a wildtype (WT, 817 bp) or knockout (KO, 386 bp) CDA1 transcript are indicated.

**Figure 3 ijms-26-02022-f003:**
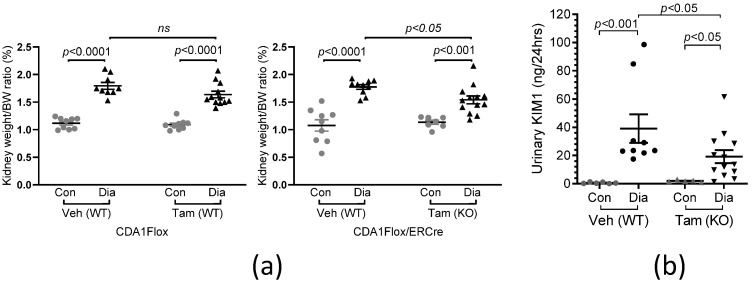
Diabetes-associated kidney hypertrophy and tubular injury are attenuated in mice with induced deletion of CDA1. (**a**) Kidney weight/body weight (BW) ratio (%) of individual mice and group mean ± SD at the end point are shown for both non-diabetic control (Con) and STZ-induced diabetic (Dia) mice of CDA1Flox (**left** panel) and CDA1Flox/ERCre (**right** panel) strains. At week 5, after induction of diabetes, these mice were treated with either vehicle (Veh) or tamoxifen (Tam), as described in the Methods. (**b**) Urinary KIM1 was measured in non-diabetic control (Con) and diabetic (Dia) CDA1Flox/ERCre mice at the end point. Individual values (ng/24 h) and group mean ± SEM are shown. *p* values between two groups (Mann–Whitney test) are shown. *ns*: statistically non-significant.

**Figure 4 ijms-26-02022-f004:**
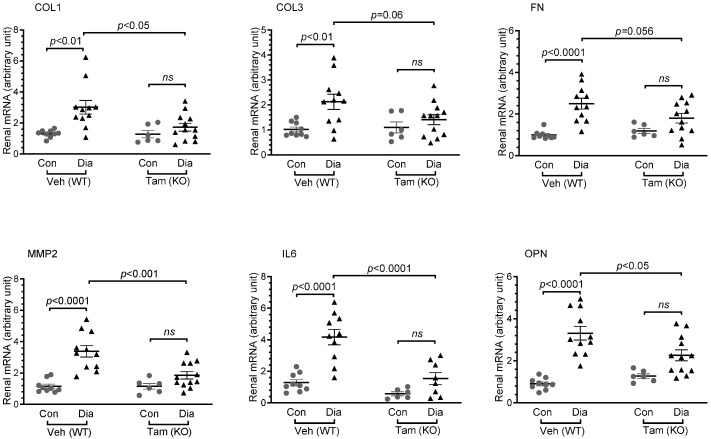
Diabetes-associated increases in renal mRNA levels of profibrotic and proinflammatory genes are attenuated in CDA1Flox/ERCre mice with induced deletion of CDA1. Renal mRNA levels of collagens I (COL1), III (COL3), fibronectin (FN), matrix metalloprotease 2 (MMP2), interleukin 6 (IL6) and osteopontin (OPN) were determined by qRT-PCR and are shown as individual values in each group, along with group mean ± SEM for CDA1Flox/ERCre mice, including non-diabetic control (Con) and diabetic mice receiving either vehicle (Veh) or tamoxifen (Tam), with the resultant genotype of either CDA1 wildtype (WT) or knockout (KO) indicated. *p* values are shown. *ns*: statistically non-significant.

**Figure 5 ijms-26-02022-f005:**
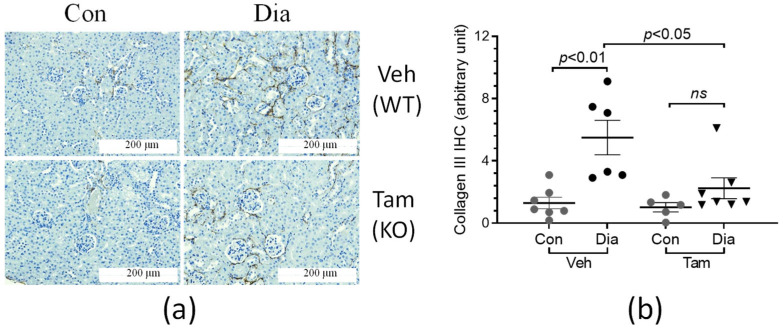
Diabetes-associated increase in renal collagen III accumulation is attenuated in CDA1Flox/ERCre mice with CDA1 deletion. (**a**) Representative microscopical images of immunohistochemical (IHC) staining of collagen III (brown color) are shown from non-diabetic control (Con) and diabetic (Dia) mice treated with either vehicle (Veh), hence having a wildtype CDA1 genotype (WT), or tamoxifen (Tam,) leading to CDA1 gene knockout (KO). (**b**) IHC staining signals were quantified by Image-Pro Plus software, Version 7.0 and arbitrary units are shown for individual values and group mean ± SEM. *p* values between specified groups are shown. *ns*: statistically non-significant.

**Figure 6 ijms-26-02022-f006:**
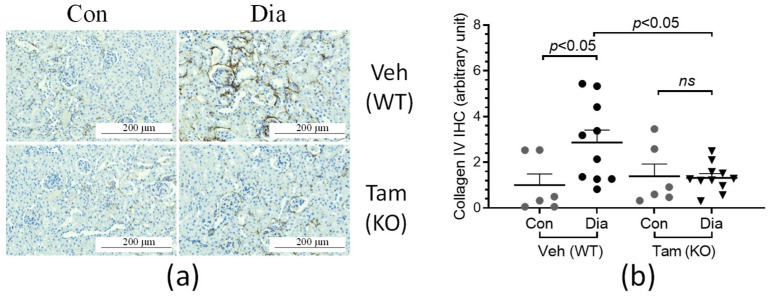
Diabetes-associated increase in renal collagen IV accumulation is attenuated in CDA1Flox/ERCre mice with CDA1 deletion. (**a**) Representative microscopical images of immunohistochemical (IHC) staining of collagen IV (brown color) are shown from non-diabetic control (Con) and diabetic (Dia) mice treated with either vehicle (Veh), hence having a wildtype CDA1 genotype (WT), or tamoxifen (Tam), leading to CDA1 gene knockout (KO). (**b**) IHC staining signals were quantified by Image-Pro Plus software, Version 7.0 and arbitrary units are shown for individual values and group mean ± SEM. *p* values between specified groups are shown. *ns*: statistically non-significant.

**Figure 7 ijms-26-02022-f007:**
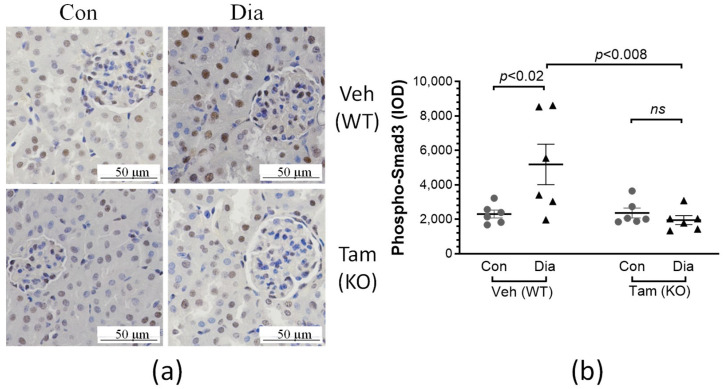
Renal TGFβ signaling is reduced in diabetic mice as a result of inducing deletion of CDA1. (**a**) Representative microscopical images of immunohistochemical (IHC) staining of phosphorylated Smad3 (Phospho-Smad3) (brown color) are shown from non-diabetic control (Con) and diabetic (Dia) mice treated with either vehicle (Veh), hence having a wildtype CDA1 genotype (WT), or tamoxifen (Tam), leading to CDA1 gene knockout (KO). (**b**) IHC staining signals were quantified by Image-Pro Plus software, Version 7.0 and calculated integrated optical density (IOD) is shown for individual values and group mean ± SEM. *p* values between specified groups are shown. *ns*: statistically non-significant.

**Table 1 ijms-26-02022-t001:** Metabolic and renal parameters of mice at week 4 of diabetes before CDA1 deletion.

Mouse Strain	CDA1Flox		CDA1Flox/ERCre	
Group	Control	Diabetic	Control	Diabetic
Body Weight (g)	30.0 ± 0.7	25.0 ± 0.4 *	30.0 ± 0.9	26.0 ± 2.2 *
Food Intake (g/day)	2.3 ± 0.1	4.8 ± 0.1 *	1.9 ± 0.2	4.5 ± 0.2 *
Water Intake (mL/day)	1.5 ± 0.2	18.4 ± 0.8 *	2.2 ± 0.2	16.0 ± 1.3 *
Urine Output (mL/day)	0.7 ± 0.1	15.5 ± 0.9 *	0.8 ± 0.1	11.4 ± 1.3 *

* *p* < 0.001 vs. corresponding non-diabetic control (*n* = 16–22).

**Table 2 ijms-26-02022-t002:** Metabolic and renal parameters of mice at the endpoint, week 10 of diabetes.

	Induction (Genotype)	CDA1Flox		CDA1Flox/ERCre	
		Control	Diabetic	Control	Diabetic
**Body Weight (g)**	vehicle (WT)	32.0 ± 1.0	24.0 ± 1.0 ***	33.0 ± 1.0	24.0 ± 1.0 ***
	Tamoxifen (KO)	33.0 ± 1.0	24.0 ± 1.0 ***	34.0 ± 1.0	26.0 ± 1.0 ***
**BG (mmol/L)**	vehicle (WT)	14.7 ± 0.9	23.1 ± 2.2 *	14.2 ± 1.2	29.5 ± 1.6 ***
	Tamoxifen (KO)	16.2 ± 0.4	25.9 ± 2.0 **	15.2 ± 1.5	24.5 ± 2.8 *
**HbA1c (%)**	vehicle (WT)	4.7 ± 0.1	11.5 ± 0.7 **	4.8 ± 0.2	12.6 ± 0.5 ***
	Tamoxifen (KO)	5.2 ± 0.3	11.2 ± 0.8 **	4.9 ± 0.2	10.6 ± 1.1 ***
**Food (g/day)**	vehicle (WT)	2.1 ± 0.4	5.8 ± 0.2 ***	1.6 ± 0.3	5.3 ± 0.2 ***
	Tamoxifen (KO)	2.9 ± 0.1	5.2 ± 0.3 *	2.1 ± 0.2	4.9 ± 0.4 ***
**Water (mL/day)**	vehicle (WT)	5.2 ± 1.6	26.5 ± 1.7 ***	3.9 ± 1.0	23.7 ± 1.2 ***
	Tamoxifen (KO)	1.8 ± 1.3	21.3 ± 2.7 ***	1.1 ± 0.3	18.7 ± 2.0 ***
**Urine (mL/day)**	vehicle (WT)	1.0 ± 0.2	23.3 ± 1.7 ***	1.1 ± 0.2	20.1 ± 1.4 ***
	Tamoxifen (KO)	1.4 ± 0.7	17.0 ± 2.5 **	1.3 ± 0.2	14.4 ± 2.8 ***

* *p* < 0.05, ** *p* < 0.01 and *** *p* < 0.001 vs. corresponding non-diabetic control (*n* = 6–13).

## Data Availability

Data contained within the article is available on request from the corresponding author.

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
