# Peer review of "Induced Genetic Deletion of Cell Division Autoantigen 1 in Adulthood Attenuates Diabetes-Associated Renal Fibrosis"

_ijms, 2025, doi:10.3390/ijms26052022_

Round 1

Reviewer 1 Report

Comments and Suggestions for Authors

Please check the attached document

Author Response

Reviewer 1

The manuscript investigates the role of CDA1 deletion in diabetic kidney disease (DKD) and its potential to attenuate renal fibrosis and inflammation. Using a tamoxifen-inducible knockout model, the study demonstrates that genetic deletion of CDA1 reduces extracellular matrix deposition and inflammation markers in diabetic mice, and the authors propose that these protective effects may be mediated through inhibition of TGFβ signaling. While the findings suggest CDA1 as a potential therapeutic target, the authors need to address the following major issues for further publication. 

Major Issues: 

  1. The authors do not provide sufficient evidence to firmly establish that CDA1 knockout attenuates renal fibrosis and inflammation via TGFβ signaling. Smad3 phosphorylation is an important part of the TGFβ pathway, but it alone does not fully support the conclusions. Other downstream targets of TGFβ or its receptors should also be evaluated by western blot or immunofluorescence staining. 

Response:

The role of CDA1 in influencing TGFβ signaling, leading to a synergistic profibrotic action in DKD has been previously established by a large body of in vitro and in vivo data from our group, including use of various global CDA1 knockout mouse strains (Tu, et al, 2011, Kidney International, 79:199-209; Chai, et al, 2013, JASN, 24:1782-92; Chai, et al, 2019, Diabetes, 68:395-408). Our previous studies have examined TGFβ ligands, TGFβ type I and type II receptors and other components of the profibrotic TGFβ pathway, as well as various inflammatory markers.

However, in the previous studies we used global CDA1 knockout mice, including CDA1/ApoE double KO mice, to validate the efficacy of genetic deletion of CDA1 in retarding DKD. These mice were born without CDA1, and were rendered diabetic in the absence of CDA1, leading to less diabetes-associated renal injury. This study focuses on characterizing the efficacy and feasibility of targeting CDA1 in a unique strain of mice, which were born with normal CDA1 gene and rendered diabetic in the presence of CDA1, a setting which is superior in terms of the clinical context where patients develop DKD in the presence of CDA1. Therefore, we consider that it is unnecessary to characterize large number of parameters of the TGFβ pathway, but we focused on the key signaling molecule Smad3 as reported in this manuscript.

In order to make this point clearer, we added some new texts in the Introduction:

“CDA1 appears to influence the TGFβ signaling pathway by increasing the expression level of the TGFβ type I receptor (TβRI), the kinase receptor directly responsible for phosphorylation and activation of the signaling molecules Smad2/3[30]. Indeed, this action of CDA1 has been shown to be specifically inhibited by the TβRI inhibitor, SB431542, as well as by the Smad3 inhibitor, SIS3[30]”

2. The description of genotyping in Figures 1 and 2 appears overly detailed and somewhat redundant. Please condense the genotyping description or move to Supplementary Data. 

Response:

We appreciate the view that there were a lot of details of the characterization data presented in Fig 1. Since this is a new mouse strain we generated for this study, we consider that the characterization of this mouse strain for the efficiency and specificity of CDA1 gene deletion induced by tamoxifen should be presented in sufficient detail. Therefore, we have largely simplified the result shown by Fig 1 and moved the genomic DNA genotyping results to a supplemental figure (Fig A1), as suggested by the reviewer.

3. The manuscript only analyzes a single 5-week time point after CDA1 deletion. The timing of CDA1 deletion (5 weeks) needs a stronger explanation. Considering the effect of CDA1 deletion within 5 weeks is limited, the author could address the long-term effects of CDA1 knockout on diabetic kidney disease (DKD). 

 Response:

Our previous study using a pharmacological approach has shown the efficacy of 5 weeks treatment of diabetic mice with a prototype CDA1 inhibitor CHA-061 (Chai, et al, 2019, Diabetes 68:395-408). Similar to the timing of CDA1 targeting in this study, the pharmacological treatment started at 6th week after STZ-induced diabetes and finished at week 10 (treatment for 5 weeks). We have added new text to make this clear in the Discussion:

  “Indeed, induced inactivation of CDA1 for 5 weeks by genetic approach in this model led to attenuation of renal parameters relevant to DKD. This is consistent with our previous findings using a pharmacological approach where STZ-diabetic mice were treated with a prototype inhibitor of CDA1, CHA-061, for 5 weeks during weeks 6-10 after diabetes induction[34].”

4. The study does not directly assess kidney function beyond fibrosis markers. Please consider adding functional kidney markers, like, urine albumin-to-creatinine ratio (UACR) or GFR measurements. 

 Response:

These issues are addressed in the Discussion section. Please see below:

“With animal models not developing extensive end-stage kidney disease[42], this study has focused on the gene expression of key molecules involved in profibrotic and proinflammatory pathways, as well as extracellular matrix accumulation, which were found to be elevated in the diabetic mice in comparison with their non-diabetic counterparts. The diabetes associated elevation of expression of these genes were attenuated as a result of absence of CDA1 for the last 5 weeks in this study, as a result of the CDA1 gene being deleted 5 weeks after induction of STZ-diabetes. Furthermore, renal accumulation of collagen III, a scarring collagen, and collagen IV, a structural basement membrane collagen, were both increased in diabetic mice and attenuated in diabetic mice with deletion of CDA1 gene. TGFβ is a profibrotic growth factor playing a key pathological role in DKD. Being consistent with our previous finding using global CDA1 knockout mice[31], TGFβ signaling assessed by the renal level of phosphorylated Smad3 in this study was significantly enhanced in diabetic mice, which was reduced to a physiological level as a result of induced CDA1 deletion.”

“Despite the attenuation of TGFβ signaling and renal fibrosis, there was no statistically significant difference of albuminuria seen between the diabetic groups with and without CDA1 (data not shown). This was, at least in part, due to a large variation of data and a relatively small group size. Furthermore, it has been known previously that mice on a C57 background do not develop robust albuminuria. Furthermore, targeting TGFβ signaling may not necessarily affect albuminuria as has been reported previously in numerous mouse models of DKD[31, 43, 44].”

5. Most quantification data are based on qPCR for gene expression, it would be better to confirm with western blot for protein level. For examples, about the CDA1 deletion validation. 

 Response:

We agree that WB is a useful quantitative method to examine protein levels in the tissue homogenate. In this study, we are interested in examining ECM accumulation, such as extracellular accumulation of collagens. We used IHC staining, which can not only quantify the proteins of interest, but also give information such as localisation of the staining signals. Similarly, we quantified the phosphorylated Smad3 as well as examining its nuclear localization in renal cells to confirm that phosphorylated Smad3 was activated by translocation from the cytoplasm to nucleus. Therefore, we consider that IHC staining used in this study produced meaningful information. As to validation of CDA1 deletion using WB, unfortunately, there are no good antibodies available to effectively examine mouse CDA1 protein in mouse kidney tissues. Furthermore, there is a single copy of CDA1 gene (Tspyl2) located to the X chromosome in the mouse genome, and we used male (XY) mice which have only one X chromosome. Therefore, we are confident that mice with genetically confirmed deletion of CDA1 do not have functional CDA1 protein.  

Minor Issues 

1. No scale bars in histology images from figure 5 and 6. 

Response:

Scale bars are added now.

2. Some of sentences are overly long and complex, the authors can rewrite them and making them easier to read. For examples, the second to last sentence in abstract. 

Response:

Manuscript has been proof read and edited.

3. There are some grammatical errors and typos, for example, in ‘funding’ section. please proofread more carefully. 

Response:

Manuscript has been proof read and edited.

4. Some figures lack clear labeling and legends, making them hard to interpret. Please figure legends clearly state what is being measured, sample sizes, and the statistical significance test used. 

Response:

Manuscript has been proof read and edited.

Reviewer 2 Report

Comments and Suggestions for Authors

Huynh P. and colleagues aimed to investigate whether the inducible genetic deletion of Cell Division Autoantigen 1 (CDA1) by using a tamoxifen-activated Cre-LoxP system in adult male diabetic mice can attenuate diabetic renal fibrosis. The results showed that tamoxifen administration significantly reduced CDA1 expression, leading to decreased TGFb signalling and lower expression of profibrotic factors. This study suggests that targeting CDA1 may be a potential therapeutic approach to mitigate the progression of diabetic nephropathy. However, some key issues remain unclear and need to be addressed.

Comments,

(1)  CDA1 knockout mice were only examined by genotype and mRNA level. CDA1 expression at the protein level by WB and IHC should be incorporated.

(2)  HbA1C (%) measurement should be included in Table 1. In addition, how many animals were analyzed, the numbers should be stated.

(3)  In Fig. 2b, there was no significant increase in CDA1 mRNA in the diabetic group (CDA1Flox/ERCre-Veh-Dia) compared to the non-diabetic group (CDA1Flox/ERCre-Veh-Con). This result is inconsistent with previous studies that CDA1 is upregulated in diabetic nephropathy (PMID: 23929772 and 20962744). Thus, in this study whether the elevation of Collagen III (Fig.5), collagen IV (Fig. 6) and p-Smad3 (Fig7) via CDA1 pathway in diabetic kidney is questionable.

(4)  In Table 2, it is not clear what’s meaning of *p<0.05, **p<0.01 and ***p<0.001 vs Control (n=6-13). CDA1Flox control or CDA1Flox/ERCre control? No statistical analysis between vehicle (WT) and Tamoxifen (KO). In addition, HbA1C was significantly lower in the diabetic-CDA1Flox/ERCre-KO group (10.6+/-1.1) compared with the diabetic-CDA1Flox/ERCre-WT (12.6+/-0.5). Thus, the diabetic-CDA1Flox/ERCre-KO group has reduced expression of collagen III (Fig. 5), collagen IV (Fig. 6), and p-Smad3 (Fig. 7), which may be due to their lower blood glucose levels but not via CDA-1 attenuation.

(5)  Can the authors explain why there is no difference (p=0.06) in COL3 mRNA (Fig. 4) but a significant difference (p<0.05) in Collagen III protein expression by IHC (Fig. 5) between Dia-Veh (WT) and Dia-Tam (KO)? Is this related to sample size? How did the authors determine the sample size to produce significant power differences?

(6)  Provide direct evidence that CDA1 KO is associated with fibrosis markers. It was necessary to double-stain CDA1 expression with collagen III (Fig. 5), collagen IV (Fig. 6), and pSmad3 (Fig. 7).

Comments on the Quality of English Language

The English could be improved to more clearly express the research.

Author Response

Reviewer 2

Comments,

1. CDA1 knockout mice were only examined by genotype and mRNA level. CDA1 expression at the protein level by WB and IHC should be incorporated.

Response:

Unfortunately, there are no good antibodies available to allow us to specifically examine mouse CDA1 protein in mouse kidney tissues. The polyclonal antibodies we generated many years ago were raised against a human CDA1 immunogen, which did not work on mouse tissues. All the commercially available antibodies we tried were not useful to detect mouse CDA1 protein specifically.    

2. HbA1C (%) measurement should be included in Table 1. In addition, how many animals were analyzed, the numbers should be stated.

Response:

Table 1 shows the metabolic data at week 4 of induced diabetes. Up to this time point, we had measured the blood glucose levels using glucometer in order to monitor the weekly blood glucose levels to ensure that the STZ-injected animals had been successfully rendered diabetic, which were then randomly allocated to be treated by either vehicle or tamoxifen. Therefore, HbA1C level was not measured at this time point. The group size is shown in the footnote: “n=16-22”. Specifically, there were >16 mice per group for non-diabetic control group of each mouse strain and 22 mice per group for diabetic mice of each mouse strain.

3. In Fig. 2b, there was no significant increase in CDA1 mRNA in the diabetic group (CDA1Flox/ERCre-Veh-Dia) compared to the non-diabetic group (CDA1Flox/ERCre-Veh-Con). This result is inconsistent with previous studies that CDA1 is upregulated in diabetic nephropathy (PMID: 23929772 and 20962744). Thus, in this study whether the elevation of Collagen III (Fig.5), collagen IV (Fig. 6) and p-Smad3 (Fig7) via CDA1 pathway in diabetic kidney is questionable.

Response:

We appreciate the comment that CDA1 gene expression was not robustly increased in diabetic mice shown in Fig 2. This was likely due to the C57/B6 background of mice used in this study. The CDA1Flox mouse strain was originally created on the C57/B6 background, which was used in this study to allow ERCre mediated inducible deletion of CDA1 gene. This mouse strain, although being useful for studying relevant signaling pathways at molecular level, has been known to be “resistant” to development of severe histological injury in the diabetic kidney. This is perhaps at least in part due to a very moderate increase in renal CDA1 expression. In our previous studies, we have shown that CDA1 expression was more robustly increased in diabetic ApoE KO mice and in diabetic rats, accompanied by more severe kidney injury. The possible links among CDA1 expression in response to diabetes, renal injury and genetic background of mice are discussed by adding a new sentence at the end of the Discussion section:

“Indeed, CDA1 gene expression was not robustly increased in diabetic kidney in this study where CDA1Flox mice and ERCre mice on C57 backgrounds were used.”

4. In Table 2, it is not clear what’s meaning of *p<0.05, **p<0.01 and ***p<0.001 vs Control (n=6-13). CDA1Flox control or CDA1Flox/ERCre control? No statistical analysis between vehicle (WT) and Tamoxifen (KO). In addition, HbA1C was significantly lower in the diabetic-CDA1Flox/ERCre-KO group (10.6+/-1.1) compared with the diabetic-CDA1Flox/ERCre-WT (12.6+/-0.5). Thus, the diabetic-CDA1Flox/ERCre-KO group has reduced expression of collagen III (Fig. 5), collagen IV (Fig. 6), and p-Smad3 (Fig. 7), which may be due to their lower blood glucose levels but not via CDA-1 attenuation.

Response:

These P values are between the diabetic and non-diabetic control groups from the same mouse strain (2 mouse strains shown). The footnote has been amended and it reads:

 “*p<0.05, **p<0.01 and ***p<0.001 vs corresponding non-diabetic Control (n=6-13)”

The difference of HbA1c levels between the diabetic CDA1Flox/ERCre KO and WT groups is not statistically significant (10.6±1.1 vs 12.6±0.5, p=0.146). All the mice in both diabetic groups have significantly higher HbA1c levels than those in the non-diabetic control groups. Therefore, we consider our diabetic model is valid and all the mice rendered diabetic had been equally exposed to the hyperglycaemic conditions.

5. Can the authors explain why there is no difference (p=0.06) in COL3 mRNA (Fig. 4) but a significant difference (p<0.05) in Collagen III protein expression by IHC (Fig. 5) between Dia-Veh (WT) and Dia-Tam (KO)? Is this related to sample size? How did the authors determine the sample size to produce significant power differences?

Response:

We agree that collagen III mRNA level failed to reach statistical significance, but showed a clear tendency towards attenuation in CDA1 KO mice. We agree that the relatively small group size could be the reason in this specific study using C57 mice which are less responsive to hyperglycaemia in terms of developing diabetes associated renal injury. We had added the following text in the Discussion to address this:

“Indeed, CDA1 gene expression was not robustly increased in the diabetic kidney in this study where CDA1Flox mice and ERCre mice on C57 backgrounds were used. This is probably reflected by some parameters failing to show statistically significant differences between the CDA1 WT and KO diabetic mice. Such an example is the mRNA level of collagen III (p=0.06) (Fig 4), although a clear tendency towards attenuation in the diabetic CDA1 KO mice was observed. Furthermore, the collagen III protein levels were shown to be significantly different in that group (p<0.05) (Fig 5). Probably, a larger group size should be considered to increase the statistical power in the future study if one uses mice on a C57 background.”

6. Provide direct evidence that CDA1 KO is associated with fibrosis markers. It was necessary to double-stain CDA1 expression with collagen III (Fig. 5), collagen IV (Fig. 6), and pSmad3 (Fig. 7).

Response:

As mentioned in our Response to the 1st comment, there are no good antibodies available to allow us to specifically examine mouse CDA1 protein in mouse kidney tissues. Thus, double staining CDA1 with other proteins cannot be performed on mouse tissues.

Round 2

Reviewer 1 Report

Comments and Suggestions for Authors

I appreciate the authors’ detailed responses and revisions based on my comments. The revisions addressed most my comments, but there are still needed extra data to address my concern about the undering mechanism about TGFb pathway.

I acknowledge that previous studies from the author’s group have extensively examined various components of the TGFβ pathway. However, since this study presents a new mouse model with inducible CDA1 deletion, additional validation of downstream signaling components could strengthen the conclusion that the protective effects are mediated through TGFβ inhibition. While the focus on Smad3 phosphorylation is reasonable, including at least one additional downstream effector (e.g., Smad2 activation, TGFβ receptor levels, or fibrotic/inflammatory markers directly regulated by TGFβ) would provide more robust support for the mechanism.

Author Response

Comment:

I appreciate the authors’ detailed responses and revisions based on my comments. The revisions addressed most my comments, but there are still needed extra data to address my concern about the undering mechanism about TGFb pathway.

I acknowledge that previous studies from the author’s group have extensively examined various components of the TGFβ pathway. However, since this study presents a new mouse model with inducible CDA1 deletion, additional validation of downstream signaling components could strengthen the conclusion that the protective effects are mediated through TGFβ inhibition. While the focus on Smad3 phosphorylation is reasonable, including at least one additional downstream effector (e.g., Smad2 activation, TGFβ receptor levels, or fibrotic/inflammatory markers directly regulated by TGFβ) would provide more robust support for the mechanism.

Response:

Thank you for your suggestion to provide more information to strengthen the conclusion that CDA1 deletion influence the TGFβ pathway by adding additional downstream effectors.

Actually, in our manuscript, we do have data to show the mRNA levels of a number of well-known TGFβ target genes, such as the fibrillar collagens I and III, fibronectin, MMP2, etc. (Fig 4). These TGFβ downstream target genes are also known to be important for fibrosis. The rise and fall of the mRNA levels of these genes reflect the effects of TGFβ/Smad3 signaling. In order to make this clearer, we have added the following context in the Discussion:

“Smad3 is phosphorylated by the TGFβ type I receptor upon activation as a result of TGFβ ligand binding, and then translocated to nucleus where it binds to the promoters and regulates the expression of TGFβ target genes. The expression levels of the TGFβ target genes examined in this study, such as collagens I, III,[44] fibronectin[45] and matrix metalloproteinase 2 (MMP2)[46] were changed in a similar way to the levels of phosphorylated Smad3 (Figs 4, 7), supporting the role of CDA1 in influencing the profibrotic TGFβ/Smad3 pathway.”

We have focused on Smad3 instead of Smad2, since Smad3 is more specifically involved in mediating the profibrotic activity of TGFβ. We added the following in the Introduction:

“TGFβ/Smad3 axis has been shown to promote renal fibrosis, whereas Smad2 is considered to counteract the profibrotic activity of Smad3.[34]”

Reviewer 2 Report

Comments and Suggestions for Authors

In the discussion, the authors should add the limitations of not having antibodies that specifically detect mouse CDA1 protein in kidney tissue by IHC and Western blotting. They should also point out that the polyclonal antibodies they generated were based on human CDA1 immunogen and did not work on mouse tissue. For the rest, I have no further questions.

Author Response

Comment:

In the discussion, the authors should add the limitations of not having antibodies that specifically detect mouse CDA1 protein in kidney tissue by IHC and Western blotting. They should also point out that the polyclonal antibodies they generated were based on human CDA1 immunogen and did not work on mouse tissue. For the rest, I have no further questions.

Response:

Thank you for your suggestion. We have now added the following text in the Discussion:

“However, CDA1 protein expression levels were not examined in this study due to lack of specific antibodies to detect mouse CDA1 in kidney tissues, which is considered to be a limitation of this study. The polyclonal antibodies we generated many years ago were raised against a human CDA1 immunogen,[11] which did not work on mouse kidney tissues.”